# Identification of risk factors and establishment of prediction models for mortality risk in patients with acute kidney injury: A retrospective cohort study

Shengtao Li[1,2], Zhanzhan Li📷[3], Yanyan Li📷[4]*

1 Department of Emergency, The First Hospital of Changsha, Changsha, Hunan Province, P.R. China,
2 Department of Emergency, Changsha Hospital, Xiangya School of Medicine, Central South University, Changsha, Hunan Province, P.R. China, 3 Department of Oncology, Xiangya Hospital, Central South University, Changsha, Hunan Province, P.R. China, 4 Department of Nursing, Xiangya Hospital, Central South University, Changsha, Hunan Province, P.R. China

* liyan4005@126.com

**Data Availability Statement:** All relevant data are within the manuscript and its Supporting information files.

## Abstract

This study investigated factors influencing death in patients with Acute Kidney Injury (AKI) and developed models to predict their mortality risk. We analyzed data from 1079 AKI patients admitted to Changsha First Hospital using a retrospective design. Patient information including demographics, medical history, lab results, and treatments were collected. Logistic regression models were built to identify risk factors and predict 90-day and 1-year mortality. The 90-day mortality rate among 1079 AKI patients was 13.8% (149/1079) and the one-year mortality rate was 14.8% (160/1079). For both 90-day and 1-year mortality in patients with AKI, age over 60, anemia, hypotension, organ failure, and an admission Scr level above 682.3 µmol/L were identified as independent risk factors through multivariate logistic regression analysis. Additionally, mechanical ventilation was associated with an increased risk of death at one year. To ensure the generalizability of the models, we employed a robust 5-fold cross-validation technique. Both the 90-day and 1-year mortality models achieved good performance, with area under the curve (AUC) values exceeding 0.8 in the training set. Importantly, the AUC values in the validation set (0.828 for 90-day and 0.796 for 1-year) confirmed that the models' accuracy holds true for unseen data. Additionally, calibration plots and decision curves supported the models' usefulness in predicting patient outcomes. The logistic regression models built using these factors effectively predicted 90-day and 1-year mortality risk. These findings can provide valuable insights for clinical risk management in AKI patients.

## Introduction

Acute kidney injury (AKI) is a potential irreversible disease characterized by a rapid decline in kidney function due to various causes, leading to multiple organ or system involvement. AKI encompasses the entire spectrum of kidney disease, ranging from mild to severe, requiring

**Funding:** The author(s) received no specific funding for this work.

**Competing interests:** The authors have declared that no competing interests exist.

renal replacement therapy [1]. The incidence of AKI in hospitalized patients is approximately 15%, while in critically ill patients, the incidence can exceed 50% [2]. Studies have shown that the incidence of AKI is gradually increasing, with over 13 million AKI patients diagnosed worldwide each year and over 1.7 million deaths attributed to AKI and its complications [3]. Globally, the incidence of AKI varies widely across different studies due to differences in definitions [4]. Previous studies have primarily focused on hospitalized patients or ICU patients, with fewer studies investigating AKI in the general population [5, 6]. However, investigating the incidence of AKI in the general population can help us understand the impact of this disease on public health.

The most prominent clinical feature of AKI is a rapid decline in kidney function, which is more common in ICU patients. Kidney damage can lead to both short- and long-term complications, such as prolonged hospitalization, renal replacement therapy, chronic kidney disease, and death [7]. Although AKI poses a significant public health burden, there are currently no effective drug prevention or treatment strategies. This may be due to the lack of effective clinical recognition of AKI, resulting in the absence of successful interventions [8]. By the time serum creatinine levels are elevated, kidney damage has already occurred. Additionally, the inability to detect subtle changes in serum creatinine makes it difficult to accurately identify AKI patients. Even after partial or complete recovery, AKI patients have a higher risk of short-term or long-term mortality.

Recent studies have reported high incidence and mortality rates for AKI, with mortality rates around 20%. As the department responsible for treating critically ill patients, the ICU has even higher incidence and mortality rates for AKI. Koyner's study found an incidence rate of 30–50% in the ICU, while Fang et al. reported an AKI mortality rate as high as 50% [9, 10]. Another study involving 54 hospitals in 23 countries showed an AKI mortality rate of 60.3% in ICU patients [11]. Based on these data, it is evident that AKI patients have a poor prognosis and high treatment costs, placing a significant burden on patients' families and society. Therefore, early detection and prevention of AKI are crucial for improving patient outcomes.

The prognosis of AKI is influenced by multiple factors. Previous studies have identified several risk factors, including age, gender, hypertension, chronic obstructive pulmonary disease, sepsis, mechanical ventilation, and organ failure [12–14]. AKI is insidious in onset, often with no early identification, warning signs, or symptoms. Continued deterioration of kidney function can be life-threatening. By utilizing these factors to construct an AKI prediction model, identifying high-risk populations, and guiding clinical treatment, we can contribute to the prevention and treatment of AKI. This study aims to analyze the risk factors for mortality in emergency AKI patients, preliminarily establish a model for predicting mortality risk, and provide a reference for clinical treatment and intervention for poor prognosis.

## Materials and methods

### Study population

This study followed the Strengthening the Reporting of Observational Studies in Epidemiology (STROBE) Statement (S1 File).

Using a retrospective cohort study, we reviewed the medical records of 1079 patients diagnosed with AKI between June 2018 and May 2020 at the First Affiliated Hospital of Changsha's Emergency Department. The patient's selection flow was presented in the Fig 1. Diagnoses were based on the 2012 KDIGO guidelines [15], defining AKI as: either a rise in serum creatinine (Scr) by ≥26.4 μmol/L within 48 hours, or an increase in Scr to ≥1.5 times baseline within a week, or urine output less than 0.5 ml/kg/h for ≥6 consecutive hours. The criteria for inclusion are as follow: (1) First hospitalization and meeting the diagnostic criteria for AKI; (2) Age range

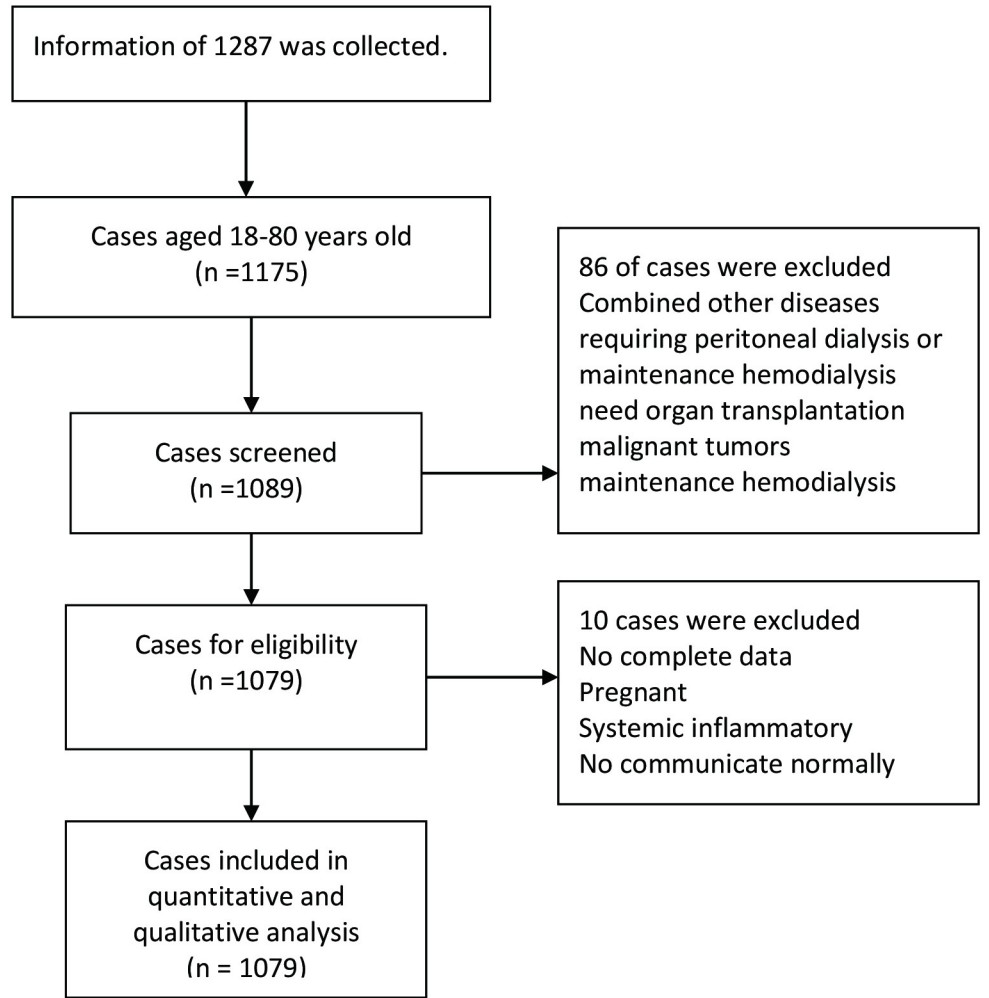

**Fig 1. The flow chart of patient's selection.**

is between 18 to 80 years old; (3) The onset to hospitalization was within 24 hours; (4) Having complete and analyzable data records; (5) For patients with multiple admissions, the data records from their first admission are considered valid. The following patients were excluded: (1) chronic kidney disease stage 5; (2) cases aged under 18 or over 80 years old; (3) patients requiring peritoneal dialysis or maintenance hemodialysis; (4) those patients who need kidney organ transplantation; patients with malignant tumors, systemic inflammatory response, or immune system diseases; (5) cases requiring nephrectomy; (6) patients who are comatose, have unclear speech, and cannot communicate normally; (7) women who are pregnant; (8) individuals with serum creatinine changes due to postrenal, glomerular, vascular, or interstitial nephritis. This study was approved by the Ethic Committee of the Xiangya Hospital of Central South University. The consent was not obtained because the data were analyzed anonymously.

## Data collection

This study determines the relevant factors based on clinical experience and references to previous research literature. The data collection was primarily obtained from medical records, and Data is

double-checked. No data missing existed. The collected data includes age(<60 vs ≥60 years old), gender (male vs female), etiological classification (renal, postrenal, and prerenal), KDIGO stage (Stage 1, 2, and 3) [16], presence of hematuria (red blood cells>3 at sediment microscopy) [17], proteinuria (urine protein ≥150 mg in 24-hour urine collection) [18], anemia (Adult males <120 g/L, adult females <110 g/L), hypertension (systolic blood pressure ≥140 mmHg and/or diastolic blood pressure ≥90 mmHg) [19], and diabetes (blood glucose ≥11.1 mmol/L or fasting blood glucose (FPG) ≥7.8 mmol/L) [20], whether mechanical ventilation was used (Acute respiratory failure syndrome caused by various etiologies requiring assisted ventilation), occurrence of hypotension (blood pressure <90/60mmHg) [21], sepsis, organ failure (Defined as dysfunction of an organ or system occurring more than 24 hours after onset), hypoalbuminemia (defined as serum albumin <30 g/L) [22], as well as the levels of hemoglobin, serum albumin, Scr, blood urea nitrogen (BUN), and blood K+ upon admission, and whether there was alternative treatment. Additionally, the study includes the outcomes for patients at 90 days and one year, including whether they were discharged, whether they died, and the time of death, along with previous medical history. The sample size is 10–20 times the number of factors studied, and the present study meet the requirements of sample size. All original data was provided in S2 File.

## Statistical analysis

Study subjects were divided into non-survivor and survivor groups based on 90-day and one-year follow-up outcomes. For normally distributed quantitative data, means ± standard deviations were used, and independent-samples t-tests were employed for comparisons between the two groups. For non-normally distributed data, Wilcoxon rank-sum tests were used. Categorical data were expressed as percentages, and chi-square tests were used for comparisons between the two groups. With 90-day and one-year mortality outcomes as dependent variables and age (≥60 vs <60), gender (male vs female), cause classification (prerenal, renal, and postrenal), KDIGO stage (1, 2, and 3), proteinuria, hematuria, anemia, hypertension, diabetes, mechanical ventilation, hypotension, sepsis, organ failure, hypoalbuminemia, admission BUN, Scr, and K+, and whether renal replacement therapy was required as independent variables, univariate and multivariate stepwise logistic regression analyses were performed to screen for relevant factors. A logistic regression model was developed to predict the 90-day and one-year mortality risk of AKI patients. The model was validated using 5-fold cross-validation with SAS software. ROC curves were used to evaluate the discrimination ability of the model in the validation and training sets, and sensitivity, specificity, and accuracy were calculated. A nomogram was used to visualize the model's predictions, and calibration curves were used to assess the model's calibration ability. Decision curves were used to evaluate the net benefit of the model's predictions for patients. $P<0.05$ was considered significant level.

## Results

### Comparisons of clinical characteristics between non-survivor and survivor in AKI patients

Fig 1 given the flow chart of patinates selection. Table 1 summarizes the clinical characteristics of non-survivors compared to survivors. The overall 90-day mortality rate was 13.8% (149/1079). Non-survivors were more likely to be older than 60 years of age (46.3% vs 24.3%, P<0.001), have a prerenal etiology of AKI (50.3% vs 31.4%, P<0.001), and anemia (31.5% vs 22.6%, P = 0.017). They also had a higher prevalence of diabetes (10.1% vs 4.5%, P = 0.005), mechanical ventilation (34.2% vs 7.5%, P<0.001), hypotension (45.0% vs 12.3%, P<0.001), sepsis (2.0% vs 0.2%, P = 0.003), and organ failure (58.4% vs 13.8%, P<0.001). Non-survivors

**Table 1. Comparison of clinical characteristics between non-survivors and survivors at 90 days and one-year.**

| Parameters | 90 days | | | One year | | |
|---|---|---|---|---|---|---|
| | Non-survivor (n = 149) | Survivor (n = 930) | P | Non-survivor (n = 160) | Survivor (n = 919) | P |
| Age (n, %) | | | <0.001 | | | <0.001 |
| ≥60 | 69(46.3) | 226(24.3) | | 74(46.3) | 221(24.0) | |
| <60 | 80(53.7) | 704(75.7) | | 86(53.8) | 698(76.0) | |
| Gender (n,%) | | | 0.207 | | | 0.473 |
| Male | 100(67.1) | 574(61.7) | | 104(65.0) | 570(62.0) | |
| Female | 49(32.9) | 356(38.3) | | 56(35.0) | 349(38.0) | |
| Etiology classification (n, %) | | | <0.001 | | | <0.001 |
| Prerenal | 75(50.3) | 292(31.4) | | 78(48.8) | 289(31.4) | |
| renal | 52(34.9) | 432(46.5) | | 58(36.3) | 426(46.4) | |
| Postrenal | 22(14.8) | 206(22.2) | | 24(15.0) | 204(22.2) | |
| KDIGO stage (n, %) | | | 0.622 | | | 0.436 |
| 1 | 4(2.7) | 34(3.7) | | 4(2.5) | 34(3.7) | |
| 2 | 10(6.7) | 79(8.5) | | 10(6.3) | 79(8.6) | |
| 3 | 135(90.6) | 817(87.8) | | 146(91.3) | 806(87.7) | |
| Albuminuria (n, %) | 61(40.9) | 430(46.2) | 0.228 | 68(42.5) | 423(46.0) | 0.408 |
| Hematuria (n, %) | 88(59.1) | 570(61.3) | 0.604 | 95(59.4) | 563(61.3) | 0.652 |
| Anemia (n, %) | 47(31.5) | 210(22.6) | 0.017 | 49(30.6) | 208(22.6) | 0.029 |
| Hypertension (n, %) | 17(11.4) | 129(13.9) | 0.415 | 17(10.6) | 129(14.0) | 0.244 |
| Diabetes (n, %) | 15(10.1) | 42(4.5) | 0.005 | 15(9.4) | 42(4.6) | 0.012 |
| Mechanical ventilation (n, %) | 51(34.2) | 70(7.5) | <0.001 | 53(33.1) | 687.4() | <0.001 |
| Hypotension (n, %) | 67(45.0) | 114(12.3) | <0.001 | 68(42.5) | 113(12.3) | <0.001 |
| Sepsis (n, %) | 3(2.0) | 3(0.2) | 0.003 | 3(1.9) | 2(0.2) | 0.004 |
| Multiple organ dysfunction (n, %) | 87(58.4) | 128(13.8%) | <0.001 | 89(55.6) | 126(13.7) | <0.001 |
| Hypoalbuminemia (n, %) | 61(40.9) | 391(42.0) | 0.800 | 63(39.4) | 389(42.) | 0.485 |
| BUN at admission, mmol/l | 31.8±14.5 | 26.3±11.8 | <0.001 | 31.3±14.2 | 26.3±11.9 | <0.001 |
| Scr at admission, mmol/l | 686.0±329.5 | 681.6±386.9 | 0.888 | 693.2±333.6 | 680.3±186.9 | 0.692 |
| $K^+$ at admission, mmol/l | 5.1±0.9 | 4.8±1.0 | <0.001 | 5.2±0.9 | 4.8±1.0 | <0.001 |
| Alternative therapy (n, %) | 89(59.7) | 521(56.0) | 0.396 | 97(60.6) | 513(55.8) | 0.258 |

also had higher initial blood potassium levels (5.1±0.9 vs 4.8±1.0, P<0.001). The two groups showed no statistically significant differences in sex, KDIGO stage 3, proteinuria, hematuria, hypertension, baseline Scr, or use of renal replacement therapy.

The one-year mortality rate for patients with AKI was 14.8% (160/1079). Non-survivors were more likely to be older than 60 years of age (46.3% vs 24.0%, P<0.001), have a prerenal etiology of AKI (48.8% vs 31.4%, P<0.001), anemia (30.6% vs 22.6%, P = 0.029), and diabetes (9.4% vs 4.6%, P = 0.012). They also had a higher prevalence of hypotension (42.5% vs 12.3%, P<0.001), sepsis (1.9% vs 0.2%, P = 0.004), and organ failure (55.6% vs 13.7%, P<0.001). Non-survivors also had higher baseline BUN (31.3±14.2 vs 26.3±11.9, P<0.001) and potassium levels (5.2±0.9 vs 4.8±1.0, P<0.001). No significant differences were observed in sex, KDIGO stage, proteinuria, hematuria, hypertension, hypoalbuminemia, or use of renal replacement therapy between the groups.

## Univariate logistic regression analysis of mortality risk in patients with AKI

Univariate analysis (Table 2) identified several risk factors for 90-day mortality in AKI patients. Patients over 60 were significantly more likely to die (OR = 2.69, 95%CI: 1.88–3.83,

**Table 2. Univariate logistic regression analysis of 90-day and one-year mortality risk in AKI patients.**

| Parameters | 90 days | | One-year | |
|---|---|---|---|---|
| | *P* | OR (95%CI) | *P* | OR (95%CI) |
| Age ($\geq$60 vs < 60) | <0.001 | 2.69(1.99–3.83) | <0.001 | 2.72(1.92–3.84) |
| Gender (Male vs Female) | 0.208 | 0.79(0.55–1.14) | 0.473 | 0.88(0.62–1.25) |
| Etiology classification | | | | |
| Renal vs prerenal | 0.001 | 2.41(1.45–4.00) | 0.001 | 2.29(1.40–3.75) |
| Prerenal vs postrenal | 0.655 | 1.13(0.67–1.91) | 0.570 | 1.16(0.70–1.92) |
| KDIGO stage | | | | |
| 2 vs 1 | 0.907 | 1.08(0.32–3.67) | 0.907 | 1.08(0.32–3.67) |
| 3 vs 2 | 0.527 | 1.41(0.49–4.02) | 0.421 | 1.54(0.54–4.40) |
| Albuminuria (Yes vs No) | 0.229 | 0.81(0.57–1.15) | 0.408 | 0.87(0.62–1.22) |
| Hematuria (Yes vs No) | 0.605 | 0.91(0.64–1.30) | 0.652 | 0.92(0.66–1.30) |
| Anemia (Yes vs No) | 0.018 | 1.58(1.08–2.31) | 0.029 | 1.51(1.04–2.19) |
| Hypertension (Yes vs No) | 0.416 | 0.80(0.47–1.37) | 0.246 | 0.73(0.43–1.25) |
| Diabetes (Yes vs No) | 0.006 | 2.37(1.28–4.39) | 0.014 | 2.16(1.17–4.00) |
| Mechanical ventilation (Yes vs No) | <0.001 | 6.40(4.22–9.70) | <0.001 | 6.20(4.11–9.36) |
| Hypotension (Yes vs No) | <0.001 | 5.85(4.01–8.53) | <0.001 | 5.27(3.64–7.63) |
| Sepsis (Yes vs No) | 0.014 | 9.53(1.58–57.55) | 0.018 | 8.76(1.45–52.85) |
| Multiple organ dysfunction (Yes vs No) | <0.001 | 8.79(6.04–12.80) | <0.001 | 7.89(5.48–11.36) |
| Hypoalbuminemia (Yes vs No) | 0.800 | 0.96(0.67–1.36) | 0.485 | 0.89(0.63–1.25) |
| BUN at admission ($\geq$27.0 vs <27.0), mmol/l | 0.061 | 1.54(0.98–2.41) | 0.033 | 1.62(1.04–2.51) |
| Scr at admission ($\geq$682.3 vs <682.3), mmol/l | 0.102 | 1.47(0.93–2.34) | 0.039 | 1.63(1.03–2.58) |
| $K^+$ at admission ($\geq$4.9 vs <4.9), mmol/l | <0.001 | 2.35(1.63–3.39) | <0.001 | 2.44(1.71–3.49) |
| Alternative therapy (Yes vs No) | 0.397 | 1.16(0.82–1.66) | 0.259 | 1.22(0.87–1.72) |

P<0.001). Compared to postrenal AKI, both renal AKI (OR = 2.41, 95%CI: 1.45–4.00, P = 0.001) and prerenal AKI (OR = 1.13, 95%CI: 0.67–1.91, P = 0.655) showed increased mortality risk, with renal AKI having a stronger association. Anemia (OR = 1.58, 95%CI: 1.08–2.31, P = 0.018) and diabetes (OR = 2.37, 95%CI: 1.28–4.39, P = 0.006) were linked to higher mortality risk. Mechanical ventilation (OR = 6.40, 95%CI: 4.22–9.70, P<0.001), hypotension (OR = 5.85, 95%CI: 4.01–8.53, P<0.001), sepsis (OR = 9.53, 95%CI: 1.58–57.55, P = 0.014), organ failure (OR = 8.79, 95%CI: 6.04–12.80, P<0.001), and high potassium levels ($\geq$4.9 mmol/L, OR = 2.35, 95%CI: 1.63–3.39, P<0.001) were all significantly associated with increased risk of death within 90 days.

No significant associations with 90-day mortality were found for sex, KDIGO stage, proteinuria, hematuria, hypertension, low albumin levels, baseline creatinine, or use of renal replacement therapy (all P>0.05).

Like the findings for 90-day mortality, univariate analysis (Table 2) revealed several risk factors for 1-year mortality in AKI patients. Patients over 60 were again significantly more likely to die (OR = 2.72, 95%CI: 1.92–3.84, P<0.001). The pattern for kidney type remained consistent, with both renal AKI (OR = 2.29, 95%CI: 1.40–3.75, P = 0.001) and prerenal AKI (OR = 1.13, 95%CI: 0.67–1.91, P = 0.655) showing no significant difference compared to postrenal AKI in terms of mortality risk. Anemia (OR = 1.51, 95%CI: 1.04–2.19, P = 0.029) and diabetes (OR = 2.16, 95%CI: 1.17–4.00, P = 0.014) remained linked to higher mortality risk. Like 90-day mortality, mechanical ventilation (OR = 6.20, 95%CI: 4.11–9.36, P<0.001), hypotension (OR = 5.27, 95%CI: 3.64–7.63, P<0.001), sepsis (OR = 8.76, 95%CI: 1.45–52.85, P = 0.018), and organ failure (OR = 7.89, 95%CI: 5.48–11.36, P<0.001) were significantly

**Table 3. Multivariate logistic regression analysis of 90-day and one-year mortality risk in AKI patients.**

| Parameters | 90 days | | One-year | |
|---|---|---|---|---|
| | **P** | **95%CI** | **P** | **95%CI** |
| Age (≥60 vs < 60) | <0.001 | 3.28(2.15–5.00) | <0.001 | 3.22(2.14–4.83) |
| Anemia (Yes vs No) | 0.017 | 1.72(1.10–2.67) | 0.026 | 1.63(1.06–2.51) |
| Hypotension (Yes vs No) | <0.001 | 4.85(30.9–7.62) | <0.001 | 4.10(2.62–6.43) |
| Multiple organ dysfunction (Yes vs No) | <0.001 | 6.02(4.00–9.07) | <0.001 | 4.48(2.87–7.00) |
| Scr at admission (≥682.3 vs <682.3), mmol/l | 0.007 | 2.13(1.23–3.68) | 0.001 | 2.42(1.41–4.16) |
| Mechanical ventilation (Yes vs No) | - | - | 0.037 | 1.75(1.04–2.97) |

associated with increased risk. Additionally, elevated levels of blood urea nitrogen (BUN), creatinine (Scr), and potassium (≥4.9 mmol/L) at admission were also risk factors for 1-year mortality (BUN: OR = 1.62, 95%CI: 1.04–2.51, P = 0.033; Scr: OR = 1.63, 95%CI: 1.03–2.58, P = 0.039; Potassium: OR = 2.44, 95%CI: 1.71–3.49, P<0.001). Like 90-day mortality, no significant associations with 1-year mortality were found for sex, KDIGO stage, proteinuria, hematuria, hypertension, low albumin levels, or use of renal replacement therapy.

## Multivariate logistic regression analysis of mortality risk in patients with AKI

According to the multivariate logistic regression analysis (Table 3), patients with AKI and the following characteristics were significantly more likely to die within 90 days: age ≥ 60 years (OR = 3.28, 95%CI: 2.15–5.00, P<0.001), anemia (OR = 1.72, 95%CI: 1.10–2.67, P = 0.017), hypotension (OR = 4.85, 95%CI: 3.09–7.62, P<0.001), organ failure (OR = 6.02, 95%CI: 4.00–9.07, P<0.001), and admission Scr level ≥682.3 μmol/L (OR = 2.13, 95%CI: 1.23–3.68, P = 0.007). These factors, along with mechanical ventilation (OR = 1.75, 95%CI: 1.04–2.97, P = 0.037), were also independent risk factors for mortality within one year.

## Development and validations of mortality risk prediction model for AKI patients

A 5-fold cross-validation method was used to build and validate the logistic regression model. The results showed that the AUC of the training set was 0.821 and the AUC of the validation set was 0.828 (Fig 2A and 2B), which were consistent. The accuracy of the training set was 88.1% and the accuracy of the validation set was 89.5%. The overall accuracy was 88.4%, the sensitivity was 77.1%, and the specificity was 73.2%. For 1-year mortality risk model, the results showed that the AUC of the training set was 0.807 and the AUC of the validation set was 0.796, which were consistent (Fig 2C and 2D). The accuracy of the training set was 87.2% and the accuracy of the validation set was 88.8%. The overall accuracy was 88.3%, the sensitivity was 76.9%, and the specificity was 69.4%.

Using training dataset, we developed Nomogram plot for predicting 90-day and one-year risk probabilities. Fig 3A and 3B show the Nomogram estimation charts for the 90-day and 1-year mortality risks of AKI patients. Taking Fig 3A as an example, if an AKI patient is over 60 years old, the age score is about 6.5 points, anemia occurs, the score is 3 points, hypotension occurs, the score is 9 points, organ failure occurs, the score is 10 points, and the Scr level is high, the score is 4.5 points, and the total score is 33 points. Projecting to Prob, the probability of death in 90 days is over 80%. The calibration plots showed that the predicted probability of nomogram fitted with the actual diagnosed nonadherences for 90-day and one-year models

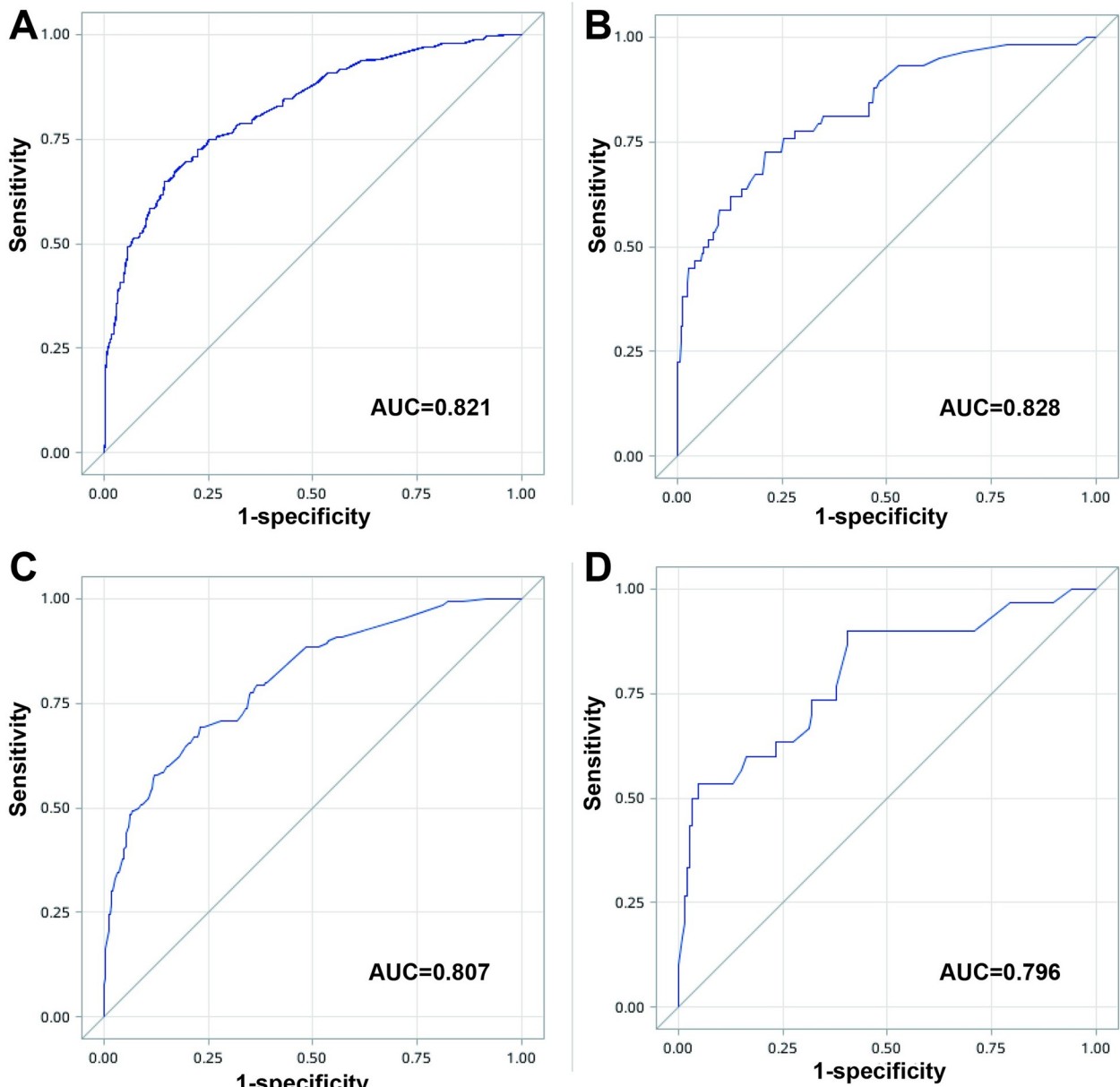

**Fig 2. Performance of the 90-day and 1-year mortality prediction models. A and B**: Receiver Operating Characteristic (ROC) curves for the training and validation sets of the 90-day prediction model. **C and D**: ROC curves for the training and validation sets of the 1-year prediction model.

(Fig 4A and 4B). The decision curves showed that patients can benefit from the prediction models (Fig 4C and 4D).

## Discussion

The results of this study indicate that the 90-day mortality rate for AKI patients is 13.8%, and the one-year mortality rate is similar at 14.8%. This mortality rate is lower than what has been reported in previous studies. Fang et al. conducted a survey of 176,155 hospitalized patients across multiple hospitals and found an incidence rate of AKI at 2.84%, with a mortality rate of

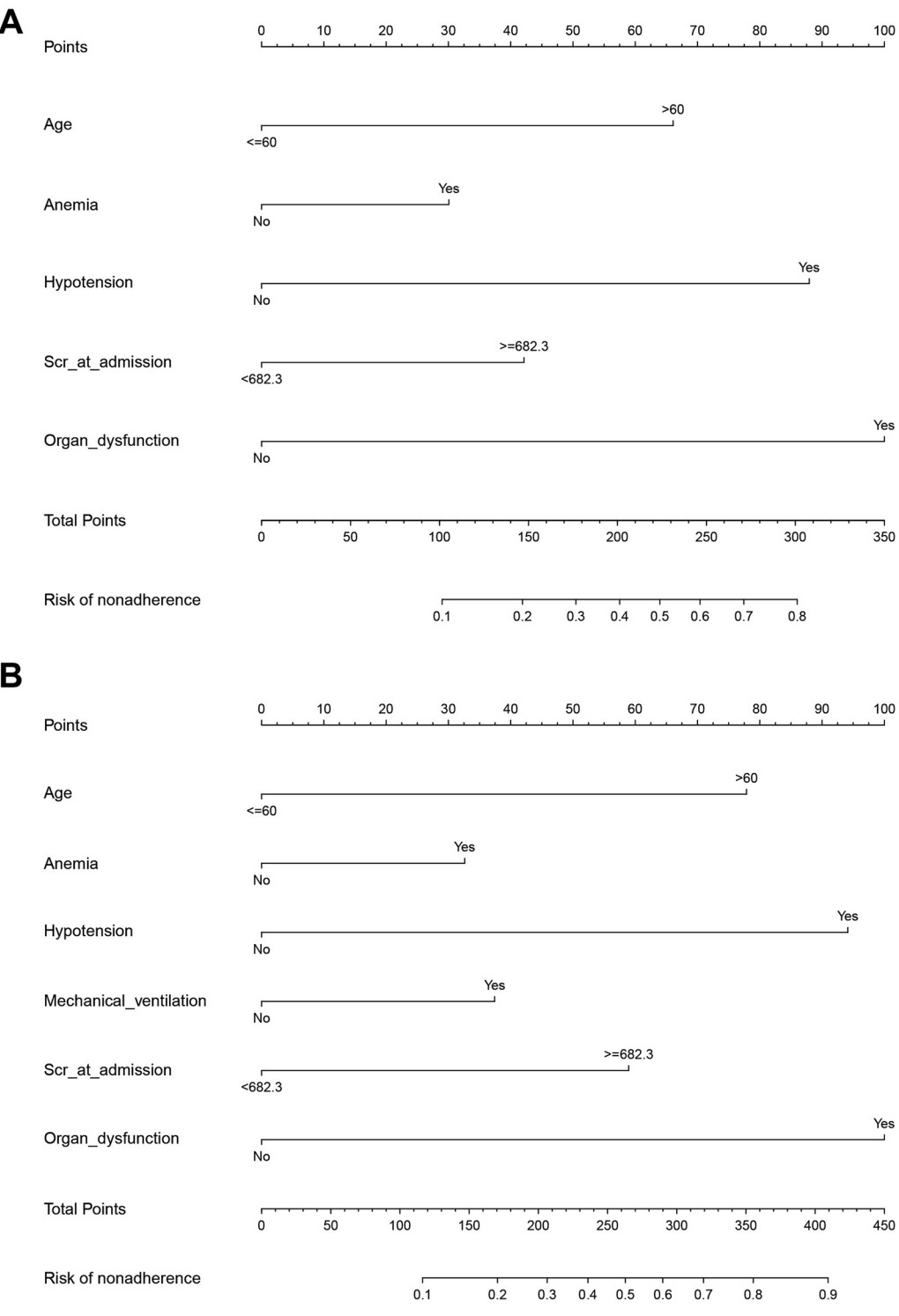

**Fig 3. Nomograms for predicting 90-day and 1-year mortality in AKI patients. A**: Nomogram for predicting the risk of death within 90 days for patients with AKI. Panel. **B**: Nomogram for predicting the risk of death within 1 year for patients with AKI.

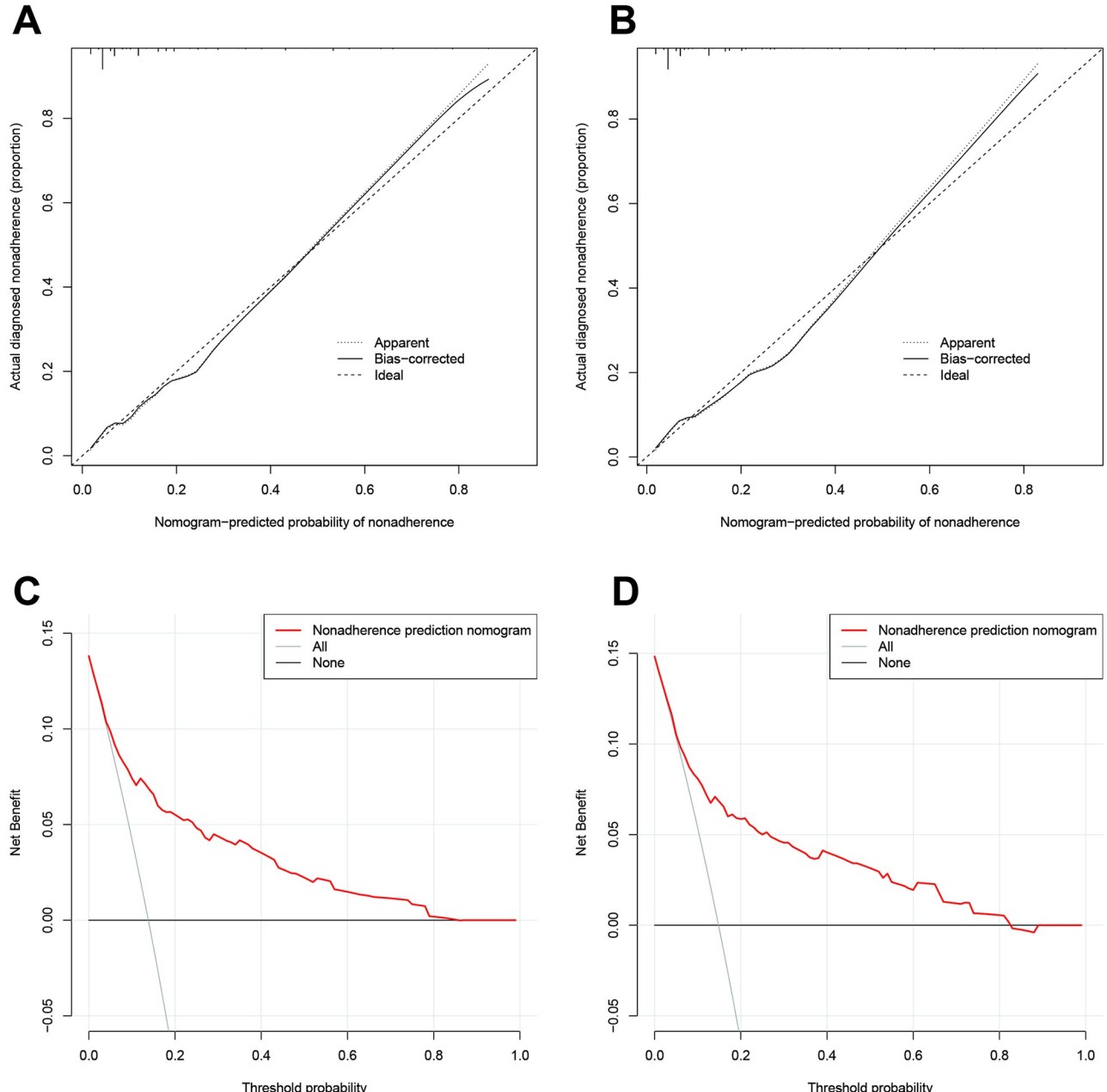

**Fig 4. Calibration and decision curves for the 90-day and 1-year mortality prediction models in AKI patients. A and B**: Calibration plots for the 90-day and 1-year prediction models, respectively. **C and D**: Decision curves for the 90-day and 1-year prediction models, respectively.

19.68% [10]. There are also reports of very high mortality rates, exceeding 50% [23]. This discrepancy may be related to the use of different diagnostic criteria and varying study conditions.

This study found that being aged 60 or older is an independent risk factor for 90-day and one-year mortality in AKI patients. First, univariate analysis indicated that the proportion of patients aged 60 or older in the mortality group was significantly higher than in the survival group. Compared to AKI patients under 60, those aged 60 or older had a 3.28-fold and

3.22-fold increased risk of 90-day and one-year mortality, respectively. Our findings are consistent with previous research. Ostermann et al. found that age is an independent risk factor for mortality in ICU patients with AKI, with the risk increasing by 2.5% per year of age [24]. The independent association between age and mortality in AKI patients may be due to the increased severity of arteriosclerosis and a gradual decline in glomerular filtration rate with age. Additionally, older patients are more prone to various complications such as hypertension and diabetes [25].

This study also found that anemia is an independent risk factor for 90-day and one-year mortality in AKI patients. Compared to non-anemic AKI patients, anemic AKI patients had a 1.72-fold and 1.63-fold increased risk of 90-day and one-year mortality, respectively. Anemia, a common condition, is associated with poor prognosis in many diseases. For example, anemia can increase the mortality risk in stroke patients. A meta-analysis of 13 cohort studies showed that anemic stroke patients had a 1.39-fold increased risk of mortality compared to non-anemic patients [26]. Among pre-dialysis CKD patients, the use of erythropoiesis-stimulating agents is rare, especially in older and more severely ill patients; these patients have high mortality and cardiovascular event rates [27]. Our study also found that organ failure is an independent risk factor for 90-day and one-year mortality in AKI patients. Compared to AKI patients without organ failure, those with organ failure had a 6.02-fold and 4.48-fold increased risk of 90-day and one-year mortality, respectively. In fact, rather than being a risk factor for AKI mortality, organ failure and AKI mortality may influence each other. Previous studies have found that organ failure can increase the mortality risk in AKI patients [28]. When AKI patients have additional organ dysfunctions, such as pneumonia, acute heart failure, or sepsis, their mortality rate can increase to 45%-60% [29].

In clinical practice, respiratory complications caused by AKI include pulmonary edema and respiratory failure requiring mechanical ventilation. Epidemiological studies have shown that respiratory failure requiring mechanical ventilation is an important independent risk factor for mortality in AKI patients [30]. This study's findings are consistent with current research, indicating that although mechanical ventilation is not related to 90-day mortality in AKI patients, it is independently associated with one-year mortality, increasing the risk by 1.75 times. Several mechanisms can explain the lung function impairment caused by AKI. Fluid retention due to AKI, resulting from reduced urine output and decreased cardiac output, is the main cause of pulmonary edema. Previous studies have considered this the primary reason for lung function impairment in AKI patients. Fluid retention and acute lung injury contribute to the increased mortality risk in AKI patients. Treating fluid overload can improve lung function and significantly shorten the duration of mechanical ventilation needed [31].

This study also found that blood pressure is independently associated with 90-day and one-year mortality risks in AKI patients. Compared to patients without hypotension, those with hypotension had a 4.85-fold and 4.10-fold increased risk of 90-day and one-year mortality, respectively. This finding is consistent with previous studies. Research indicates that hypotension is not only a risk factor for the occurrence of AKI but also an independent risk factor for mortality in AKI. Abu et al. found that low systolic blood pressure at admission is independently associated with mortality risk in 319 elderly AKI patients [32]. Decreased blood pressure reduces organ perfusion, including renal blood flow, leading to multiple organ failures, including the kidneys. The severity and duration of hypotension are closely related to the occurrence of AKI. Studies have shown that when the mean arterial pressure is below 80 mmHg, the risk of AKI increases by 3% for every mmHg decrease in blood pressure; if hypotension persists for more than 10 hours, the ischemic damage to the kidneys becomes irreversible [33]. Additionally, this study found that the serum creatinine (Scr) level at admission is a risk factor for 90-day and one-year mortality in AKI patients. Scr is the most direct indicator

of renal function, and high Scr levels indicate the extent of kidney damage in AKI patients, reflecting poor prognosis. When Scr levels rise, it signifies severe kidney damage, but it only becomes significantly elevated when the glomerular filtration rate decreases to one-third of the normal rate. At this point, Scr itself becomes a harmful substance, leading to hyperuricemia, indirectly demonstrating the relationship between admission Scr levels and AKI patient prognosis.

Currently, there are many predictive models for the occurrence of AKI, but fewer models predict AKI prognosis. Previous studies have attempted to predict AKI mortality. Zhou in China explored the influencing factors of pregnancy-related AKI prognosis and established a predictive model [34]. This study used serological indicators to construct a logistic regression model, achieving a sensitivity of 0.85, specificity of 1.0, and an accuracy of 0.977. However, the study did not validate the model, had a small sample size, which may lead to overfitting, and did not use mortality as the study outcome. Liu Zhang et al. used a Cox regression model to predict the 28-day mortality risk in surgery-related AKI patients, identifying several potential indicators through serological markers [35]. The logistic regression model constructed in this study predicts the 90-day and one-year mortality risks in AKI patients, achieving good predictive performance in both the training and validation sets. Decision curve analysis shows that patients can benefit from the predictive model.

This study has certain limitations. Firstly, the subjects included in this study were primarily emergency patients, who may be more critically ill than general inpatients. Therefore, the population setting should be considered when applying these findings. This also led to some differences in risk factors affecting mortality between our study and previous studies. Secondly, this study aimed to construct a simple and practical model, thus only clinical symptoms were included as study factors, excluding serological indicators. Including some serological indicators might further improve the predictive ability of the model. Additionally, this study only predicted the 90-day and one-year mortality risk, indicating the necessity of long-term follow-up. Future studies should include survival data with follow-up time to potentially construct better models. Finally, the sample size in this study was limited, necessitating further studies with larger sample sizes. These studies should also conduct stratified predictions for different populations and follow-up outcomes to enhance the practicality of the constructed model.

In this study, 90-day and one-year mortality rates of patients with acute AKI were relatively high. Our study indicated that age $\geq$ 60 years, anemia, symptoms of hypotension, organ failure, admission Scr level $\geq$ 682.3, organ failure, and mechanical ventilation are independent factors associated with 90-day or one-year mortality in AKI patients. The Logistic regression model constructed based on these factors can effectively predict the 90-day and one-year mortality risk for AKI patients. Our study will provide theoretical reference for clinical risk management.

## Supporting information

**S1 File. STROBE checklist.**
(DOC)

**S2 File. Original data.**
(XLS)

## Author Contributions

**Conceptualization:** Shengtao Li.

**Data curation:** Shengtao Li, Yanyan Li.

**Formal analysis:** Zhanzhan Li.

**Investigation:** Yanyan Li.

**Methodology:** Zhanzhan Li.

**Software:** Zhanzhan Li.

**Supervision:** Yanyan Li.

**Validation:** Zhanzhan Li.

**Visualization:** Shengtao Li.

**Writing – original draft:** Shengtao Li.

**Writing – review & editing:** Zhanzhan Li, Yanyan Li.

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
