## [Decision Letter · Decision Letter 0]

3 Sep 2024

PONE-D-24-24695Identification of risk factors and establishment of prediction models for mortality risk in patients with acute kidney injury: A retrospective cohort studyPLOS ONE

Dear Dr. Li,

Thank you for submitting your manuscript to PLOS ONE. After careful consideration, we feel that it has merit but does not fully meet PLOS ONE’s publication criteria as it currently stands. Therefore, we invite you to submit a revised version of the manuscript that addresses the points raised during the review process.

We appreciate your study which is an interesting study. However, there are some points raised by the reviewer and need to be clarified. Please carefully respond to the reviewer comments and suggestions.

We look forward to receiving your revised manuscript.

Kind regards,

Vipa Thanachartwet, M.D.

Academic Editor

PLOS ONE

Journal Requirements:

Reviewers' comments:

Reviewer's Responses to Questions

**Comments to the Author**

1. Is the manuscript technically sound, and do the data support the conclusions?

Reviewer #1: Partly

2. Has the statistical analysis been performed appropriately and rigorously? 

Reviewer #1: Yes

3. Have the authors made all data underlying the findings in their manuscript fully available?

Reviewer #1: Yes

4. Is the manuscript presented in an intelligible fashion and written in standard English?

Reviewer #1: Yes

5. Review Comments to the Author

Reviewer #1: This is a single retrospective study try to explore 2 focal targets as following

1. Demonstrating independent factors influencing the 90 day and 1 year mortality in patients with Acute Kidney Injury (AKI) among 1079 non surgical related AKI patients admitted to Changsha First Hospital, China during June 2018 and May 2020.

2. Developed and validate models to predict the above patient mortality risk.

My comments and concerns:

1. Feng et al. In JAMA Network Open 2023;6:e2313359, has presented an article entitled “ Characterization of Risk Prediction Models for Acute Kidney Injury: A Systematic Review and Meta-analysis”. In addition, Feng developed a prediction model for AKI using a statistical model included 150 studies from literature search containing 14.4 million participants. Moreover, well known epidemiological data regarding AKI which have been reported elsewhere in current textbook and current literatures ((UptoDate 2024, Liangos et al in “Kidney and patient outcomes after acute kidney injury in adults”).

So this is not an original one to report the results related to the purpose of the manuscript. In addition, this manuscript report restrospective data from a single center with subjects of 1079 patients which is much smaller comparing to the JAMA report.

2. The authors develop a model to generate a mortality risk prediction model and validate the model utilize the 5-fold cross validation technique. However, it is not cleared about how the normogram in Figure 3 are derived from.

3. The authors in this manuscript conclude age ≥ 60 years, anemia, symptoms of hypotension, organ failure, admission Scr level ≥ 682.3 micromoles/L, organ failure, and mechanical ventilation are independent factors associated with 90-day or one-year mortality in AKI and the Logistic regression model constructed based on these factors can effectively predict the 90-day and one-year mortality risk for AKI patients. In addition they conclude their study will provides theoretical reference for clinical risk management.

It has been known that factors influencing mortality have been described elsewhere (UptoDate 2024, Liangos et al in “Kidney and patient outcomes after acute kidney injury in adults”). In the mentioned review, Liangos report odd ratio of the factors influencing mortality. In various AKI reports have mention additional factors such as post surgery, underlying cardiovascular or cerebrovascular events, APACHE scores

criteria of the study.

6. PLOS authors have the option to publish the peer review history of their article (what does this mean?). If published, this will include your full peer review and any attached files.

Reviewer #1: No

---

## [Author Response · Author response to Decision Letter 0]

19 Sep 2024

Response to Editorial office

Comment 1: Please ensure that your manuscript meets PLOS ONE's style requirements, including those for file naming. The PLOS ONE style templates can be found at 

Response 1: Yes, we have revied the format of manuscript according to these requirements. 

Comment 2: Please include your full ethics statement in the ‘Methods’ section of your manuscript file. In your statement, please include the full name of the IRB or ethics committee who approved or waived your study, as well as whether you obtained informed written or verbal consent. If consent was waived for your study, please include this information in your statement as well. 

Response 2: Yes, we have provided this. 

Comment 3: Please include captions for your Supporting Information files at the end of your manuscript, and update any in-text citations to match accordingly. Please see our Supporting Information guidelines for more information: http://journals.plos.org/plosone/s/supporting-information. 

Response 3: Yes, we have revises this. 

Response to Reviewer #1: 

This is a single retrospective study try to explore 2 focal targets as following

1. Demonstrating independent factors influencing the 90 day and 1 year mortality in patients with Acute Kidney Injury (AKI) among 1079 non-surgical related AKI patients admitted to Changsha First Hospital, China during June 2018 and May 2020.

2. Developed and validate models to predict the above patient mortality risk.

My comments and concerns:

Comment 1: Feng et al. In JAMA Network Open 2023;6:e2313359, has presented an article entitled “ Characterization of Risk Prediction Models for Acute Kidney Injury: A Systematic Review and Meta-analysis”. In addition, Feng developed a prediction model for AKI using a statistical model included 150 studies from literature search containing 14.4 million participants. Moreover, well known epidemiological data regarding AKI which have been reported elsewhere in current textbook and current literatures ((UptoDate 2024, Liangos et al in “Kidney and patient outcomes after acute kidney injury in adults”). So, this is not an original one to report the results related to the purpose of the manuscript. In addition, this manuscript reports retrospective data from a single center with subjects’ of 1079 patients which is much smaller comparing to the JAMA report.

Response 1: Thank you for your advice. We totally agree with you that some studies about this topic had been published. First, this study is totally different from ours. This study focusses one the occurrence of AKI, and our study is to evaluate the prognosis of AKI. Different study population in different region may have different results, even small differences. Plos one is a very inclusive journal, that the editors of Plos one makes decisions on submissions based on scientific rigor, That the editors of Plos one makes decisions on submissions based on scientific rigor, regardless of novelty. It is precisely because of this that we choose to contribute to this journal, which provides us with a good platform for publishing research, so that even small researchers like us can make their own voices. According to the statistical experience, the sample size should be is 10-20 times the number of factors studied, and the present study meet the requirements of sample size.

Comment 2: The authors develop a model to generate a mortality risk prediction model and validate the model utilize the 5-fold cross validation technique. However, it is not cleared about how the nomogram in Figure 3 are derived from.

Response 2: Thank you for your advice. The nomogram in Figure 3 was based on the training dataset. We performed the multivariate logistic regressions and only included these variables in the nomogram that were significant in the model. this is routing data analyses strategy. We have added some descriptions in the results. 

Comment 3: The authors in this manuscript conclude age ≥ 60 years, anemia, symptoms of hypotension, organ failure, admission Scr level ≥ 682.3 micromoles/L, organ failure, and mechanical ventilation are independent factors associated with 90-day or one-year mortality in AKI and the Logistic regression model constructed based on these factors can effectively predict the 90-day and one-year mortality risk for AKI patients. In addition, they conclude their study will provides theoretical reference for clinical risk management. It has been known that factors influencing mortality have been described elsewhere (UptoDate 2024, Liangos et al in “Kidney and patient outcomes after acute kidney injury in adults”). In the mentioned review, Liangos report odd ratio of the factors influencing mortality. In various AKI reports have mention additional factors such as post-surgery, underlying cardiovascular or cerebrovascular events, APACHE scores criteria of the study.

Response 3: Thank you for your advice. We read the publication: Characterization of Risk Prediction Models for Acute Kidney Injury: A Systematic Review and Meta-analysis. You can see this study is a systematic review and meta-analysis, which to systematically review published AKI prediction models across all clinical sub settings. This study is totally different from ours. This study focusses one the occurrence of AKI, and our study is to evaluate the prognosis of AKI. The study outcomes is completely different. Though the sample size is very large, and it seems our study data is higher efficiency. Moreover, this study also indicated that the variation in the clinical settings, populations, and predictive variables are wide, which makes these study high statistical heterogeneity that could not be ascribed to any particular variable other than geographical region. This means study population in different regions may have different characteristics. Our study also contributed some results among these models. The UptoDate seems to need some fee that we can’t afford, which is difference between us and other higher-level hospital. You also mentioned that in various AKI reports have mention additional factors such as post-surgery, underlying cardiovascular or cerebrovascular events, APACHE scores criteria of the study. I think this may be due to different study population, clinical management way, treatment (different treatment may have different medical data), and hospital level. But anyway, the study could be appropriate when the data was applied in local region based on current situation. We also added some descriptions in discussion.

---

## [Decision Letter · Decision Letter 1]

8 Oct 2024

Identification of risk factors and establishment of prediction models for mortality risk in patients with acute kidney injury: A retrospective cohort study

PONE-D-24-24695R1

Dear Dr. Li Yanyan,

We’re pleased to inform you that your manuscript has been judged scientifically suitable for publication and will be formally accepted for publication once it meets all outstanding technical requirements.

Kind regards,

Vipa Thanachartwet, M.D.

Academic Editor

PLOS ONE

Additional Editor Comments (optional):

All issues have been addressed.

Reviewers' comments:

Reviewer's Responses to Questions

**Comments to the Author**

1. If the authors have adequately addressed your comments raised in a previous round of review and you feel that this manuscript is now acceptable for publication, you may indicate that here to bypass the “Comments to the Author” section, enter your conflict of interest statement in the “Confidential to Editor” section, and submit your "Accept" recommendation.

Reviewer #1: All comments have been addressed

2. Is the manuscript technically sound, and do the data support the conclusions?

Reviewer #1: Yes

3. Has the statistical analysis been performed appropriately and rigorously? 

Reviewer #1: Yes

4. Have the authors made all data underlying the findings in their manuscript fully available?

Reviewer #1: Yes

5. Is the manuscript presented in an intelligible fashion and written in standard English?

Reviewer #1: Yes

6. Review Comments to the Author

Reviewer #1: None

7. PLOS authors have the option to publish the peer review history of their article (what does this mean?). If published, this will include your full peer review and any attached files.

Reviewer #1: **Yes: **Thanom Supaporn MD. PhraMongkutklao Hospital and College of Medicine, Bangkok, Thailand

---

## [Editor Report · Acceptance letter]

11 Oct 2024

PONE-D-24-24695R1 

PLOS ONE

Dear Dr. Li, 

I'm pleased to inform you that your manuscript has been deemed suitable for publication in PLOS ONE. Congratulations! Your manuscript is now being handed over to our production team.

Kind regards, 

on behalf of

Professor Vipa Thanachartwet 

Academic Editor

PLOS ONE